# Tracking of Internal Granular Progenitors Responding to Valproic Acid in the Cerebellar Cortex of Infant Ferrets

**DOI:** 10.3390/cells13040308

**Published:** 2024-02-07

**Authors:** Shiori Kamiya, Tetsuya Kobayashi, Kazuhiko Sawada

**Affiliations:** 1Division of Life Science, Graduate School of Science and Engineering, Saitama University, Saitama 338-8570, Japan; sk.dietitian2017@gmail.com (S.K.); tkoba@mail.saitama-u.ac.jp (T.K.); 2Department of Nutrition, Faculty of Medical and Health Sciences, Tsukuba International University, Tsuchiura 300-0051, Japan

**Keywords:** cerebellum, VPA, neurogenesis, Bergmann glia, ferret

## Abstract

Internal granular progenitors (IGPs) in the developing cerebellar cortex of ferrets differentiate towards neural and glial lineages. The present study tracked IGPs that proliferated in response to valproic acid (VPA) to determine their fate during cerebellar cortical histogenesis. Ferret kits were used to administer VPA (200 μg/g body weight) on postnatal days 6 and 7. EdU and BrdU were injected on postnatal days 5 and 7, respectively, to label the post-proliferative and proliferating cells when exposed to VPA. At postnatal day 20, when the external granule layer was most expanded, EdU- and BrdU-single-labeled cells were significantly denser in the inner granular layer of VPA-exposed ferrets than in controls. No EdU- or BrdU-labeling was found in Purkinje cells and molecular layer interneurons. Significantly higher percentages of NeuN and Pax6 immunostaining in VPA-exposed ferrets revealed VPA-induced differentiation of IGPs towards granular neurons in BrdU-single-labeled cells. In contrast, both EdU- and BrdU-single-labeled cells exhibited significantly greater percentages of PCNA immunostaining, which appeared in immature Bergman glia, in the internal granular layer of VPA-exposed ferrets. These findings suggest that VPA affects the proliferation of IGPs to induce differentiative division towards granular neurons as well as post-proliferative IGPs toward differentiation into Bergmann glia.

## 1. Introduction

Cerebellar neurons originate from progenitor pools in the upper rhombic lip (uRL) and ventricular zone of the roof of the fourth ventricle of the embryonic cerebellum. External granular precursors (EGPs) arise from the uRL at embryonic day 12 or 13 [1] and subsequently migrate to surround the cerebellar surface entirely, forming the external granular layer (EGL) [2]. The cerebellar cortex consists of the following three layers: the molecular layer (ML): rich parallel fibers (axons of granular neurons) and Purkinje cell dendrites and scattering in basket and stellate neurons (ML interneurons); the Purkinje cell layer (PCL): composed of soma of Purkinje cells and Bergmann glia aligned in a monolayer; and the inner granular layer (IGL): densely packed with granular neurons and sparsely populated excitatory and inhibitory interneurons such as unipolar brush cells, Golgi cells, and Lugaro cells [3]. In mice, EGP proliferation peaks around postnatal day (PD) 8 and then EGPs migrate radially along Bergmann glial processes while differentiating into internal granular neurons, forming the IGL by PD 20 [1,4,5,6].

Ferrets, belonging to the order Carnivora, undergo a similar trajectory of cerebellar cortical histogenesis to rodents [7,8]. Recently, we found progenitors, named internal granule progenitors (IGPs), in the IGL of the ferret cerebellum, which are not seen in the mouse cerebellum [9,10]. IGPs have potency to differentiate into both neural and glial linages in the developing cerebellar cortex of newborn ferrets [10]. Valproic acid (VPA), an anticonvulsant/antiepileptic drug and an inhibitor of histone deacetylases 1 and 2 [11], facilitated the neuronal differentiation of IGPs following their proliferation [10]. While IGPs may be another source of granular neurons distinct from EGPs [10], the glial cell types differentiated from IGPs were unclear. Furthermore, our previous study did not elucidate the possibility of IGP differentiation into other cortical neurons such as ML interneurons [10]. In this study, IGPs were tracked to determine the neuronal and glial cell types that differentiate, particularly in response to VPA. This study was an extension of a previous study [10], in which VPA was injected into infant ferrets following the same schedule on PDs 6 and 7 when the EGL began to expand [8]. This study used the cerebellar cortex of ferrets at PD 20, when the EGL is most expanded with three distinguishable cortical layers. Using the ferret cerebellar cortex at this age allows the stages of granular neurogenesis to be distinguished by immunohistochemistry [9].

## 2. Materials and Methods

### 2.1. Animals

Eight naturally-delivered male pups from six pregnant ferrets, which were purchased from Japan SLC (Hamamatsu, Japan), were used in the experiments. The pups were reared with lactating dams (3–5 pups/mother) in stainless steel cages (80 cm × 50 cm × 35 cm) maintained at 21.5 ± 2.5 °C under 12-h artificial illumination in the Facility of Animal Breeding, Nakaizu Laboratory Japan SLC (Izu, Japan). All lactating dams were fed a pellet feed (High-Density Ferret Diet 5L14, PMI Feeds, Inc., St. Louis, MO, USA) and ad libitum tap water.

The thymidine analogs and VPA were administered on a schedule designed according to a previous study [10]. Eight male pups were intraperitoneally injected with 30 μg/g body weight of 5-ethynyl-2′-deoxyuridine (EdU) (Sigma-Aldrich, St. Louis, MO, USA) at PD 5 and 30 μg/g body weight of 5-bromo-2′-deoxyuridine (BrdU) (Sigma-Aldrich) at PD 7, respectively. Four pups were intraperitoneally administered 200 µg/g body weight of VPA on PDs 6 and 7. Four unexposed pups were used as controls. VPA was injected a second time at the same time as the BrdU injection. On PD 20, all animals were perfused with 4% paraformaldehyde (PFA) (Merck, Darmstadt, Germany)-phosphate buffered saline (PBS) (pH 7.4) under deep anesthesia with ~2% isoflurane gas. The cerebellum was removed and immersed in the same fixative.

### 2.2. Preparation of Tissue Sections

Each cerebellum was soaked overnight in PBS containing 30% sucrose and then embedded in an optimal cutting temperature compound (Sakura Finetech Japan, Tokyo, Japan) at −70 °C. After embedding, 40-μm-thick sagittal cryosections of the cerebellum were made using a Retratome (REM-700: Yamato Koki, Asagiri, Japan) equipped with an electro freeze (MC-802A, Yamato Koki). All sections were collected in vials containing 4% PFA-PBS.

### 2.3. Immunohistochemical Procedures

Six serial sections (100 μm thickness, no gaps between adjacent sections) were prepared from the cerebellar vermis around the midsagittal plane. One section was stained with hematoxylin for gross histological evaluation. The remaining five sections were used for EdU and BrdU detections with immunostaining for five different antigens. All immunostaining procedures were carried out on floating sections. The sections were heated in the Antigen Retrieval Reagent UNIVERSAL (R&B system, Minneapolis, MN, USA) at 90 °C for 30 min in a water bath and then cooled at 4 °C for 30 min, followed by preincubation in PBS containing 0.1% TritonX-100 (Triton-PBS) at 37 °C for 1 h. The Click-iT EdU Alexa Fluor Imaging Kit (Thermo Fisher Scientific, Waltham, MA, USA) was used for the detection of EdU. Then, the sections were reacted with the primary antibodies diluted in Triton-PBS containing 10% normal horse serum (Funakoshi, Tokyo, Japan) at 4 °C overnight. Primary antibodies used are shown in Appendix A. These were highly specific for ferret brain tissue [8,9]. Then, sections were incubated at 37 °C for 2 h with a mixture of appropriate combinations of secondary antibodies listed in Appendix A. When biotinylated anti-mouse IgG was used as a secondary antibody, sections were further reacted with Alexa555-labeled streptavidin (1:500; S21381, Thermo Fisher Scientific) at 37 °C for 2 h. Here, the terms “positive labeling” or “immunopositive” (+) refer to cells labeled with thymidine analogs or marker antigens at high and medium levels, and “no labeling” or “immunonegative” (−) to cells labeled at low levels or unlabeled cells.

### 2.4. Evaluation of Cell Density

Serial digital sectioning images were acquired at a 10 μm depth (1 μm section plane thickness, 10 sections) from the most superficial plane, and the locations of immunostained cells with EdU and/or BrdU labeling were obtained. Images were acquired using an Axio Imager M2 ApoTome.2 with a 20× objective lens, equipped with an AxioCam MRm camera (Zeiss, Gottingen, Germany) and the Zen 2.3 blue edition software (Zeiss). A set of sectional images, 4 μm apart in the Z-direction (third and seventh from the superficial slices of the acquired images), were selected as the reference and lookup images, respectively. The density of thymidine analogs or immunostained cells was estimated by the disector method using systematic random sampling following a previously established procedure [10]. In each section, 6 square box frames (box size = 40 × 40 µm) were used to systematically select the region of interest (ROI) randomly placed on the ML, PCL, and IGL of vermal lobules III, V, VI, and VIII of both the reference and lookup images, at the same positions perpendicular to the surface. Thymidine analog-labeled or immunostained cells were counted within the ROIs using the “forbidden line” rule [12]. Their densities were calculated using the following formula: [Cell density = Qn−/(a × b × t)] (Qn− = total number of thymidine analog-labeled and/or immunostained cells appearing within ROIs in the lookup images, but not in the reference images; a = 6, total number of ROIs in the lookup images per animal; b = 40 μm × 40 μm areas of counting frame; and t = 4 μm, distance between the lookup and reference images). Densities of thymidine analog-labeled or immunostained cells in each cortical layer were estimated by summing measurements from lobules III and V as the anterior lobe and from lobule VI and VIII as the posterior lobe.

The percentage of immunostained cells among thymidine analog-labeled cells was estimated by summing the number of cells counted within all ROIs from all animals in each group. Because the percentage was estimated independently for each marker antigen, it was unclear their overlaps.

### 2.5. Statistical Treatment

The cell densities of thymidine analog-labeled and immunostained cells in the ML, PCL and IGL were evaluated statistically using a repeated-measures two-way analysis of variance (ANOVA) with the cerebellar lobes (anterior and posterior) and groups (VPA-exposed and control) as factors. Scheffe’s test was carried out as post-hoc analysis following the significant differences in group and/or region × group interactions detected by the repeated-measures two-way ANOVA and simple main effects. The percentage of cells immunostained for marker antibodies in EdU+ or BrdU+ cells was evaluated statistically using the χ^2^ test, with “n” defined as the total count of EdU+ and BrdU+ cells.

## 3. Results

### 3.1. Gross Structures of the Cerebellum

The counterclockwise torque asymmetry of the cerebellum was distinct in adult male ferrets [13]. The left/right-side bias of the gross morphology of the cerebellum was not seen on PD 20 in the VPA-exposed or control group (Figure 1a,b). Figure 1c shows the hematoxylin-stained midsagittal sections of the cerebellar vermis of VPA-exposed and control groups. The cerebellar vermis was highly lobulated, allowing ten lobules to be distinguished with no difference in their gross morphology between groups.

### 3.2. Densities of EdU- and BrdU-Labeled Cells

EdU and/or BrdU labeling appeared in neuronal and glial linages derived from IGPs as well as EGP-derived granular neurons when thymidine analogs were administered to ferret pups on the same schedule as in the present study [10]. We tracked IGP-derived neurons and glial cells on PD 20 when the EGL was most expanded [8]. Distributions of thymidine analog-labeling and/or immunostained cells in the cerebellar cortex were not optically different among the vermal lobules examined, i.e., lobules III, V, VI, and VIII in both VPA-exposed and control groups. Figure 2, Figure 3 and Figure 4 depict representative immunofluorescence staining for various markers with EdU and BrdU labelings in the lobule VIII cortex adjacent to the secondary fissure.

EdU was labeled in cells proliferated on PD 5, 24 h prior to the first VPA administration. EdU+ cells were sparsely distributed throughout the ML, PCL, and IGL, but not the EGL, in both VPA-exposed and control ferrets on PD 20 (Figure 2a). BrdU was labeled in cells proliferated on PD 7, immediately following the second VPA administration. BrdU+ cells were distributed sparsely in the ML through PCL and diffusely in the IGL, with those in the IGL more abundant in the VPA-exposed group (Figure 2a). No BrdU+ cells were observed in the EGL in the VPA-exposed or control group (Figure 2a). Throughout the cerebellar cortical layers, EdU+/BrdU+ cells, which underwent two rounds of cell division at a 48-h interval, were rarely found in both groups (Figure 2a).

The densities of EdU+, BrdU+- and EdU+/BrdU+ cells were estimated in each cerebellar cortical layer of the anterior and posterior lobes of the vermis. Repeated-measures two-way ANOVA revealed a significant effect of VPA exposure [*F*_(1,6)_ = 6.5271; *p* < 0.05] on the EdU+ cell density. Scheffe’s test indicated significantly denser EdU+ cells in the IGL of the anterior lobe in the VPA-exposed group than in the control group (*p* < 0.05) (Figure 2b). The ANOVA results did not indicate a significant effect on cerebellar lobe × group interaction, showing that the difference in the EdU+ cell density between two groups was not cerebellar lobe-specific. Therefore, similar to the anterior lobe, the EdU+ cell density in the IGL tended to be higher in the posterior lobe of VPA-exposed ferrets.

The density of BrdU+ cells was significantly higher in the PCL and IGL of both the anterior and posterior lobes of the VPA-exposed group than in the control group by Scheffe’s test (Figure 2b). Repeated-measures two-way ANOVA revealed the same differences: a significant effect on a VPA exposure in the PCL [*F*_(1,6)_ = 10.9521; *p* < 0.05] and in the IGL [*F*_(1,6)_ = 12.0981; *p* < 0.05]. The EdU+/BrdU+ cell density in any cortical layers examined did not differ in the anterior and posterior lobes between the VPA-exposed and control groups (Figure 2b).

### 3.3. Double Immunofluorescence Staining for Calbindin D28k and Parvalbumin with EdU and BrdU Labeling

Calbindin D28k is a specific marker for Purkinje cells, and parvalbumin is expressed selectively in Purkinje cells and ML interneurons such as basket and stellate neurons in the cerebellum [14]. We used antibodies against these antigens to detect EdU and BrdU labeling in Purkinje cells (calbindin D28k+/parvalbumin+) and ML interneurons (parvalbumin+). EdU and BrdU labelings were not found in calbindin D28k+/parvalbumin+ Purkinje cells and parvalbumin+ ML interneurons in VPA-exposed and control groups (Figure 2a). Thus, IGPs proliferated on PDs 5 and 7 may not differentiate into Purkinje cells or ML interneurons, and VPA did not lead IGPs to differentiate towards these cortical neurons.

### 3.4. Immunofluorescence Staining for Various Markers with EdU and BrdU Labeling

EdU and BrdU labeling was carried out with immunofluorescence staining for various marker antigens, such as paired box 6 (Pax6), neuronal nuclear antigen (NeuN), proliferating cell nuclear antigen (PCNA), S100, and brain fatty acid binding protein type 7 (BLBP). The overlapping expression of these marker antigens was unclear because we estimated the percentages independently. In both VPA-exposed and control groups, few or no EdU+/BrdU+ cells were found in any cortical layers examined so that the immunostaining ratios for each antigen in these cells were not estimated. Furthermore, there were no EdU+ and BrdU+ cells in the EGL of both VPA-exposed and control groups (Figure 2a). This analysis was not performed in the EGL.

#### 3.4.1. Ratio of Pax6 Immunostaining

Pax6 is expressed in migrating/differentiating granular neurons and IGPs in the developing cerebellum [9,15]. Many BrdU+ cells were immunostained for Pax6 in both VPA-exposed and control groups (Figure 3a). The ratio of Pax6 immunostaining in BrdU+ cells was significantly higher in the ML of the VPA-exposed group in the anterior lobe, but not in the posterior lobe (Table 1). This difference in the Pax6 immunostaining ratio suggests an earlier onset of the differentiation of EGPs into granular neurons in the VPA-exposed group than in the control group. In contrast, in the posterior lobe, the ratio of Pax6 immunostaining in BrdU+ cells was 100% in the PCL and 94.3% in the IGL of the VPA-exposed group, significantly higher than in the PCL (62.5%) and IGL (79.1%) of the control group (Table 2 and Table 3). The onset of neuronal differentiation from IGPs, which proliferated on PD 7, may be also earlier in the VPA-exposed group than in the control group during cerebellar corticohistogenesis. The percentage of Pax6 immunostaining in EdU+ cells did not differ between two groups in any cortical layers of the anterior and posterior lobes (Table 1, Table 2 and Table 3).

#### 3.4.2. Ratio of NeuN Immunostaining

NeuN is a mature neuronal marker that is expressed in granular neurons but absent in other cerebellar cortical neurons such as Purkinje cells and ML interneurons [16,17]. NeuN+ granular neurons were primarily found in the IGL (Figure 3b). The presence of a small number of NeuN+ cells in the ML through PCL suggests that EGP-derived migrating/differentiating granular neurons began to express NeuN. A significantly lower NeuN immunostaining ratio was detected in BrdU+ cells in the ML of the posterior lobe and in EdU+ cells of the PCL of the anterior lobe in VPA-exposed ferrets than in control ferrets (Table 1 and Table 2). VPA-affected EGPs may facilitate Pax6 expression in their migrating/differentiating state (see the above subsection) but delay the onset of NeuN expression, resulting in slow maturation. In contrast, the percentage of NeuN immunostaining in BrdU+ cells was significantly higher in the IGL of the posterior lobe of VPA-exposed group than in the control group (Table 3). Similar to the findings of Pax6 immunostaining, NeuN immunostaining also revealed that the maturation of IGP-derived granular neurons was facilitated by exposing proliferating IGPs to VPA.

#### 3.4.3. Ratio of PCNA Immunostaining

PCNA+ cells were abundant in the EGL and IGL adjacent to the PCL in both VPA-exposed and control groups (Figure 4). In the IGL, the ratio of PCNA immunostaining was significantly higher in EdU+ cells in the anterior lobe and in BrdU+ cells in both anterior and posterior lobes of VPA-exposed group than in the control group (Table 3). However, the PCNA immunostaining ratio in both EdU+ and BrdU+ cells did not differ between the groups in the ML in the PCL (Table 1 and Table 2). PCNA immunostaining overlapped with S100 immunostaining, which appeared mainly in differentiating Bergmann glia located in the IGL adjacent to the PCL (Appendix A). Therefore, the differentiated glial linage from IGPs may be Bergmann glia. Furthermore, Bergmann glia differentiation may be facilitated by exposing proliferating and/or post-proliferative IGPs to VPA.

#### 3.4.4. Ratio of S100 Immunostaining

The S100 gene was expressed largely in immature Bergman glia in the developing mouse cerebellum from embryonic day13.5 to PD 3 [18]. S100+ cells were found in the IGL adjacent to the PCL in the ferret cerebellum on PD 7 [10], and some of them were co-stained with anti-PCNA on PD 20 (Appendix A). Therefore, S100 may be expressed in differentiating/immature Bergmann glia as well as mature Bergman glia in the ferret cerebellar cortex. S100 immunostaining appeared mainly in Bergmann glia located in the PCL and underlying IGL in the ferret cerebellar cortex on PD 20 (Figure 5a). Mature Bergmann glia aligned in the PCL by PD 42 [8]. The ratio of S100 immunostaining in BrdU+ cells was significantly higher in the PCL but not the IGL of both anterior and posterior lobes in VPA-exposed group (Table 2). An increased density of BrdU+ cells (see Figure 2b,c) with S100 expression in the PCL of the VPA-exposed group suggests the enhanced migration of Bergmann glia derived from VPA-affected IGPs into the PCL. (Table 1, Table 2 and Table 3). The S100 immunostaining ratio in EdU+ cells did not differ between VPA-exposed and control groups in three cortical layers examined (Table 2).

#### 3.4.5. Ratio of BLBP Immunostaining

BLBP is a differentiation marker for Bergmann glia in the mouse cerebellum [19]. In the ferret cerebellar cortex, BLBP immunostaining was also found in S100+ Bergmann glia on PD 20 and thereafter (Appendix A). BLBP immunostaining appeared mainly in the PCL and the IGL adjacent to the PCL of the ferret cerebellar cortex on PD 20 (Figure 5b). The percentage of BLBP immunostaining in BrdU+ cells was significantly higher in the IGL of the anterior lobe, but not in other cortical layers of the anterior and posterior lobes, of the VPA-exposed group than in the control group (Table 1, Table 2 and Table 3). Thus, VPA may promote BLBP expression-defined maturation following IGP differentiation into Bergmann glia. Conversely, BLBP immunostaining was rarely found in EdU+ cells distributed in any cortical layers in both groups (Table 1, Table 2 and Table 3).

### 3.5. Densities of Cells Immunostained for Various Marker Antigens

Figure 6 shows the total densities of cells immunostained for each marker antigen in the cerebellar cortex of VPA-exposed and control groups on PD 20. A significant effect on groups was detected in the PCNA+ [*F*_(1,6)_ = 8.0761; *p* < 0.05] and the S100+ cell densities [*F*_(1,6)_ = 9.8467; *p* < 0.05] in the PCL by the repeated-measures two-way ANOVA. Scheffe’s test indicated significantly greater PCNA+ and S100+ cell densities in the PCL, but not in the ML and IGL, of the posterior lobe of VPA-exposed and control groups (Figure 6). The ANOVA results did not indicate a significant effect on cerebellar lobe × group interaction, showing that the differences in PCNA+ and S100+ cell densities in the PCL between two groups were not cerebellar lobe-specific. Therefore, similar to the posterior lobe, PCNA+ and S100+ cell densities in the PCL tended to be higher in the anterior lobe of VPA-exposed ferrets. The results support the findings mentioned above that IGP-derived Bergmann glia migrate into the PCL more in VPA-exposed ferrets than in control groups.

## 4. Discussion

The present study tracked IGP-derived cells, particularly those which differentiate more in response to VPA, to determine their cell fate during cerebellar cortical histogenesis. On PD 20, when the EGL was most expanded in the ferret cerebellar cortex [8], granular neurons and Bergmann glia were mainly tracked from IGPs proliferated on PDs 5 and 7. Tracked cells in the ML were immunopositive for NeuN but not for parvalbumin, suggesting that ML interneurons are not derived from IGPs. Therefore, IGPs may be another source of granular neurons distinct from EGPs in the ferret cerebellar cortex. Furthermore, VPA prevented the proliferation of EGPs on PD 7 [10] and facilitated the maturation of IGP-derived granular neurons with a delayed maturation of EGP-derived granular neurons on PD 20. The cerebellar volume was estimated using MR images obtained from brain samples of ferrets exposed to VPA on the same administration schedule as in a previous study [20]. The cerebellar volume did not differ between VPA-exposed and control ferrets (Appendix A), suggesting that VPA facilitates replacement of the main source of granular neurons from EGPs to IGPs during cerebellar cortical histogenesis in ferrets. Such changes associated with VPA exposure may not involve the loss of EGPs, IGPs or their derived cells. VPA enhanced anti-apoptotic factors (such as Bcl-2) in neural progenitors [21,22] and inhibited apoptotic cell death [23].

It is unclear whether the contributions to the cerebellar function of EGP- and IGP-derived granular neurons differ. However, an increased ratio of IGP-derived granular neurons in the IGL may alter cerebellar function, because neonatal VPA exposure causes autism spectrum disorder-like social behavior deficits in ferrets [24]. Further investigation should focus on the differences between IGP- and EGP-derived granular neurons with respect to their connections to other neurons including cerebellar cortical interneurons and cerebellar afferents such as climbing and mossy fibers. IGPs also differentiated into Bergmann glia in addition to granular neurons. An enhanced density with increasing PCNA immunostaining ratio in EdU+ and BrdU+ cells was observed in the IGL of the VPA-exposed group on PD 20. The differentiating Bergmann glia were PCNA immunopositive and abundant in the IGL adjacent to PCL (Appendix A). Our previous study revealed that PCNA expression in post-proliferative (EdU+) IGPs was sustained immediately following VPA exposure on PD 7 [10]. Because of the neurogenic-to-gliogenic transition, neural stem cells, such as apical radial glia involved in cerebral corticohistogenesis, undergo a neurogenic phase followed by a gliogenic phase [25]. Therefore, the sustained undifferentiated state of post-proliferative IGPs by VPA exposure may result in accumulating the IGP population that differentiate into Bergmann glia. Furthermore, the facilitated maturation of Bergmann glia derived from VPA-affected IGPs was revealed by BLBP expression and their migration to the PCL.

Bergmann glia are generated from the ventricular zone of the roof of the fourth ventricle [26] that gives rise to deep cerebellar neurons, Purkinje cells, and progenitors of ML interneurons (prospective white matter) in mouse fetuses at mid-gestation [2]. It is unclear whether IGPs are identical to progenitors of classically known Bergmann glia in mice. From another perspective, EGPs, another origin of granular neurons distinct from IGPs in ferrets, arise from the uRL [3,6]. The developmental and evolutionary significances of IGPs are intriguing if IGPs and EGPs originate from different germinal zones, that is, germinal epithelia of the fourth ventricle and the uRL, respectively. Further investigation is needed whether IGPs are present in various mammalian species.

In the cerebellar cortex of adult mice, Sox2-expressing progenitors (SEP, also known as nestin-expressing progenitors) are arranged in the PCL with Bergmann glia, and are involved in neurogenesis following cerebellar injury and motor learning [19]. Notably, SEPs lack S100 expression [19] that may be crucial for sustaining neurogenic potency by preventing astroglial differentiation, similar to hippocampal radial glial cells for adult neurogenesis [27]. S100-negative SEP-like cells were also found in the PCL and underlying IGL in the cerebellar cortex of young adult ferrets [8]. We need to investigate whether SEPs emerge from IGPs as a derivative of Bergmann glia by modifying expressions of glia-specific proteins such as S100.

## 5. Conclusions

Recently, we reported the presence of a progenitor, named IGP, that had potency to differentiate both into neural and glial linages in the developing cerebellar cortex of ferrets [10]. In the current investigation, IGPs proliferated on PD 5 or 7 differentiated into granular neurons and Bergmann glia. VPA administration experiments further suggest epigenetic regulation of IGP differentiation towards these two derivatives. While IGPs may be a secondary source of granular neurons distinct from EGPs in the ferret cerebellar cortex, neonatal VPA exposure increased IGP-derived granular neurons but decreased EGP-derived granular neurons in relative neuronal populations composing the IGL. In humans, the cerebellum plays roles in working memory, spatial attention, and cognition in addition to its classically known functions such as the regulation of muscle tone and the integration of sensory and motor functions [28,29,30]. Investigating the existence of IGPs in other mammals, particularly in higher order primates such as humans, will provide an understanding of the evolutionary significance of granular neurons in the diversity of cerebellar function among mammalian species.

## Figures and Tables

**Figure 1 cells-13-00308-f001:**
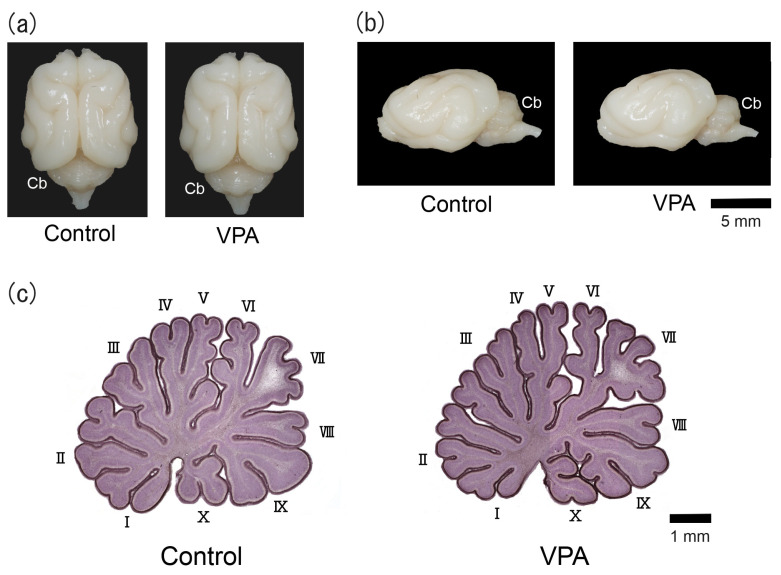
Optical images of VPA-exposed and control ferrets on postnatal day 20. (**a**) Dorsal view of the brains. (**b**) Lateral view of the left side of the brains. (**c**) Hematoxylin-stained midsagittal sections of the cerebellar vermis. Roman numerals are used to identify the vermal lobules. Cb, cerebellum.

**Figure 2 cells-13-00308-f002:**
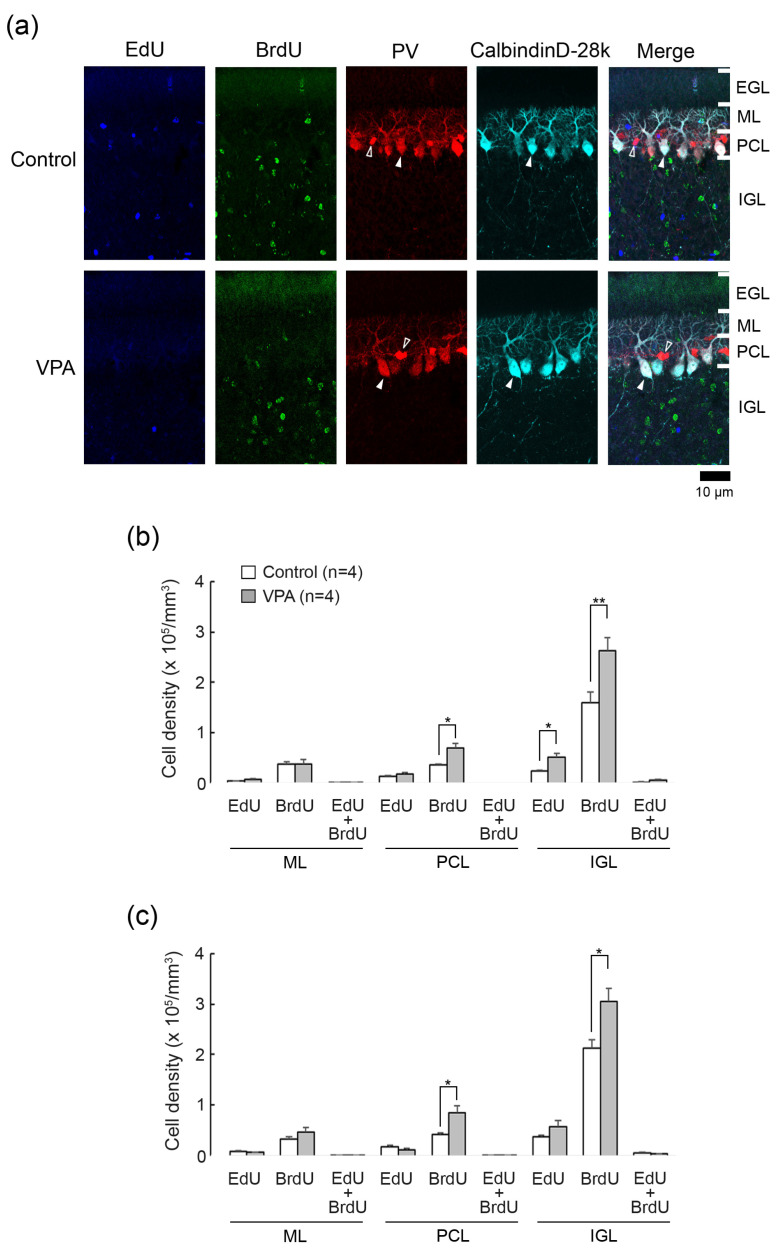
Double immunofluorescence staining for calbindin D28k and parvalbumin with EdU and BrdU labeling in cerebellar cortex of VPA-exposed and control ferrets on postnatal day 20. (**a**) Double immunofluorescence staining for calbindin D28k and parvalbumin (PV) with EdU and BrdU labeling in lobule VIII of the vermis. Open arrowheads, PV immunopositive molecular layer interneurons; Closed arrowheads, calbindin D28k/PV-double-positive Purkinje cells. No EdU- and BrdU labeling appeared in Purkinje cells and molecular layer interneurons. (**b**) Bar graphs indicating densities of EdU-single-, BrdU-single- and EdU/BrdU-double-labeled cells in the cerebellar cortex of the anterior lobe. * *p* < 0.05, ** *p* < 0.01 (Scheffe’s test). (**c**) Bar graphs indicating density of EdU-single-, BrdU-single- and EdU/BrdU-double-labeled cells in the cerebellar cortex of the posterior lobes. * *p* < 0.05 (Scheffe’s test) Data in (**b**,**c**) represent mean ± standard error. EGL external granular layer; IGL, inner granular layer; ML, molecular layer; PCL, Purkinje cell layer.

**Figure 3 cells-13-00308-f003:**
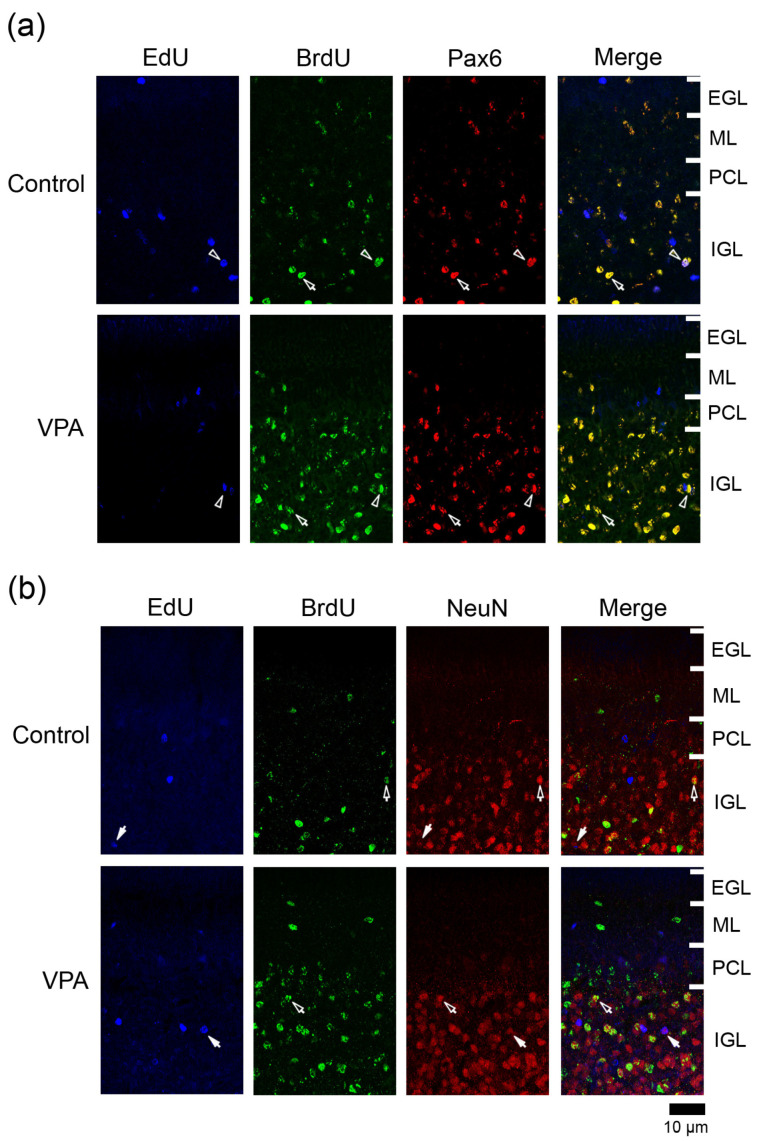
Immunofluorescence staining for Pax6 or NeuN with EdU and BrdU labeling in cerebellar cortex of VPA-exposed and control ferret on postnatal day 20. (**a**) Pax6 immunofluorescence staining with EdU and BrdU labeling in the lobule VIII of the vermis. Open arrowheads, EdU/BrdU-double-labeled Pax6 immunopositive cells; arrows, BrdU-single-labeled Pax6 immunopositive cells. (**b**) NeuN immunofluorescence staining with EdU and BrdU labeling in lobule VIII of the vermis. Open arrows, BrdU-single-labeled NeuN immunopositive cells. Closed arrows, EdU-single-labeled NeuN-immunopositive cells. EGL external granular layer; IGL, inner granular layer; ML, molecular layer; PCL, Purkinje cell layer.

**Figure 4 cells-13-00308-f004:**
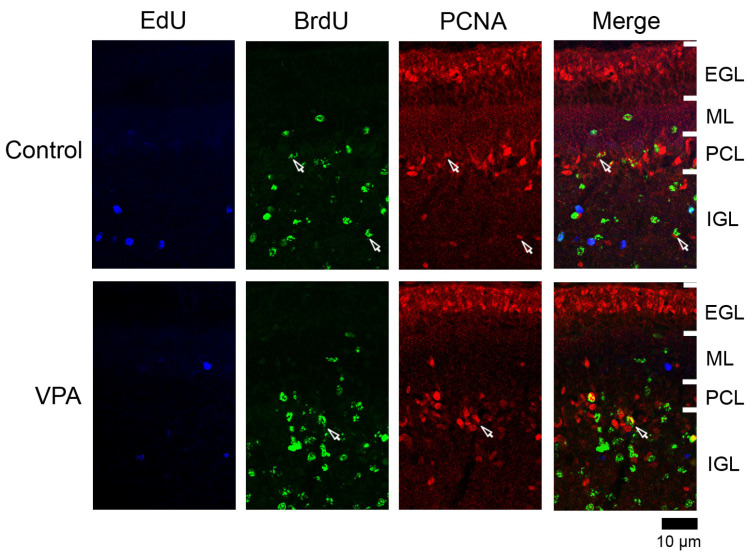
PCNA immunofluorescence staining with EdU and BrdU labeling in cerebellar cortex of VPA-exposed and control ferrets on postnatal day 20. Fluorescence-stained images in lobule VIII of the vermis are shown. Open arrows, BrdU-single-labeled PCNA immunopositive cells. EGL external granular layer; IGL, inner granular layer; ML, molecular layer; PCL, Purkinje cell layer.

**Figure 5 cells-13-00308-f005:**
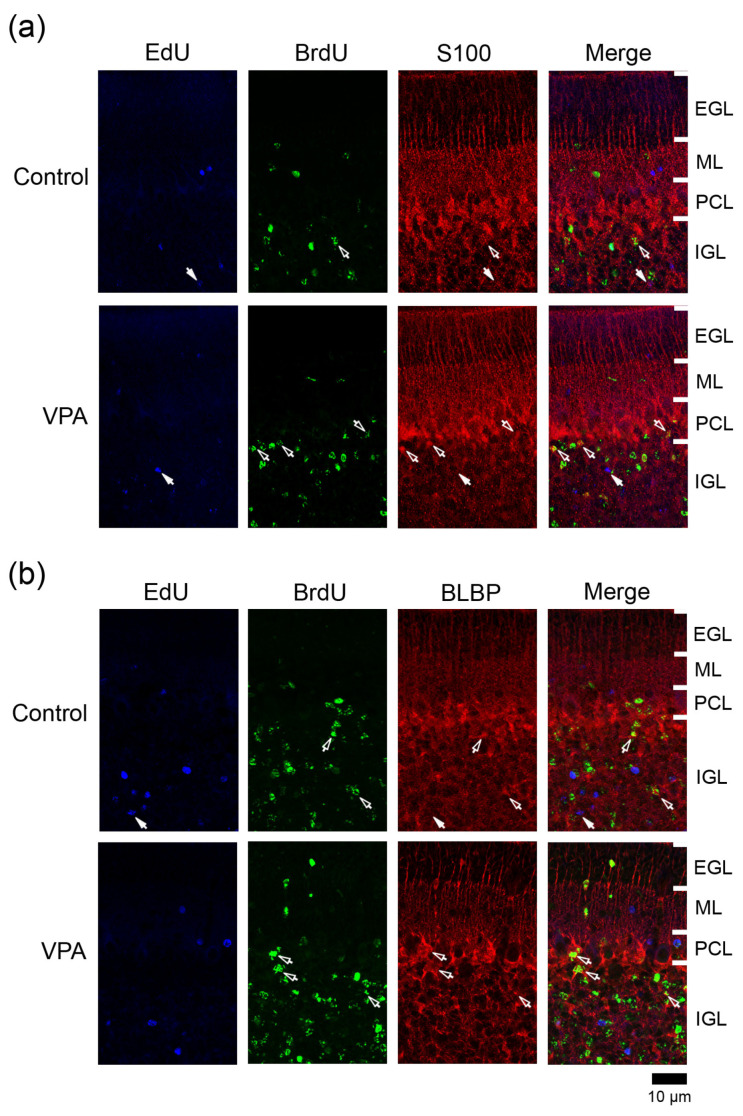
Immunofluorescence staining for S100 or BLBP with EdU and BrdU labeling in the cerebellar cortex of VPA-exposed and control ferrets on postnatal day 20. (**a**) S100 immunofluorescence staining with EdU and BrdU labeling in lobule VIII of the vermis. Open arrows, BrdU-single-labeled S100 immunopositive cells; Closed arrows, EdU-single-labeled S100 immunopositive cells. (**b**) BLBP immunofluorescence staining with EdU and BrdU labeling in lobule VIII of the vermis. Open arrows, BrdU-single-labeled BLBP immunopositive cells; Closed arrows, EdU-single-labeled BLBP immunopositive cells. EGL external granular layer; IGL, inner granular layer; ML, molecular layer; PCL, Purkinje cell layer.

**Figure 6 cells-13-00308-f006:**
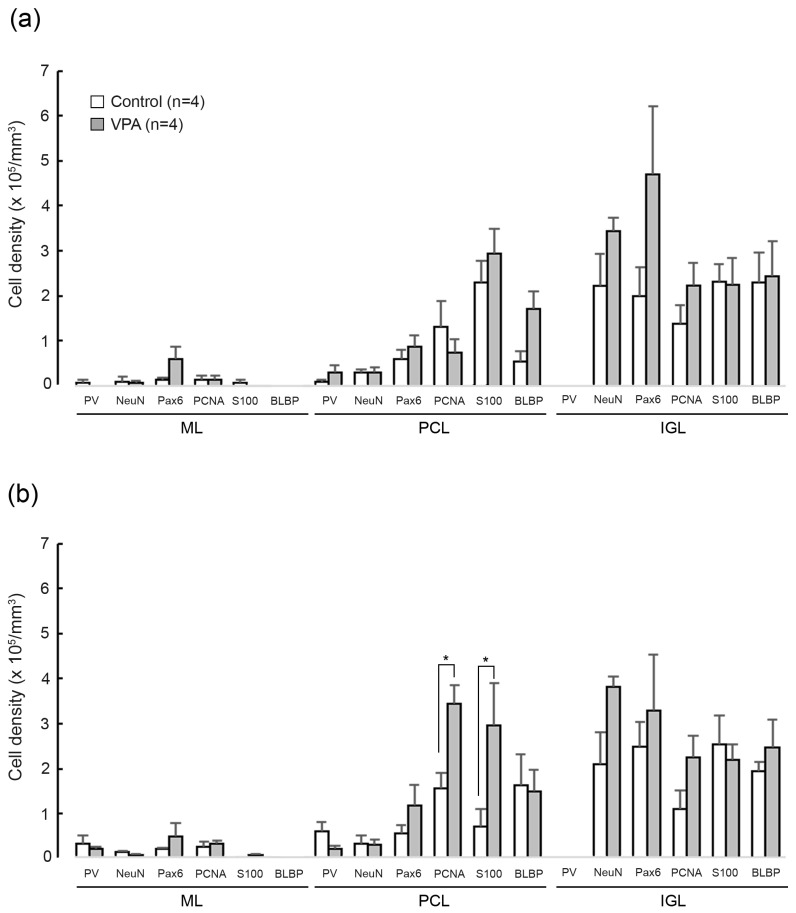
Bar graphs indicating densities of cells immunostained for various marker antigens in cerebellar cortical layers of VPA-exposed and control ferrets on postnatal day 20. (**a**) Anterior lobe. (**b**) Posterior lobe. Data represent means ± standard error. * *p* < 0.05 (Scheffe’s test). IGL, inner granular layer; ML, molecular layer; PCL, Purkinje cell layer.

**Table 1 cells-13-00308-t001:** Percentages of immunostained cells for various markers in EdU-single- and BrdU-single-labeled cells in the molecular layer of the cerebellar cortex.

	Anterior Lobe		Posterior Lobe	
	Control	VPA	Control	VPA
EdU-single labeled cells				
% of Pax6	0% (0/1)	0% (0/3)	100% (1/1)	25% (1/4)
% of NeuN	33.3% (1/3)	50.0% (1/2)	0% (0/3)	50.0% (1/2)
% of PCNA	0% (0/4)	0% (0/5)	0% (0/4)	0% (0/1)
% of S100	ND	0% (0/2)	0% (0/4)	0% (0/3)
% of BLBP	0% (0/2)	0% (0/3)	0% (0/4)	0% (0/4)
BrdU-single labeled cells				
% of Pax6	50.0% (2/4)	92.9% (13/14) *	83.3% (5/6)	100% (8/8)
% of NeuN	14.3% (1/7)	8.3% (1/12)	60.0% (3/5)	8.3% (1/12) *
% of PCNA	0% (0/23)	0% (0/13)	0% (0/20)	6.3% (1/16)
% of S100	11.8% (2/17)	0% (0/15)	0% (0/17)	0% (0/15)
% of BLBP	0% (0/17)	0% (0/18)	0% (0/9)	0% (0/28)

Percentages are calculated by summing each labeled cell counted within all ROIs from the molecular layer of the cerebellar cortex from four ferrets. The number of each labeled cell for calculating the percentages is shown in parentheses. ** p* < 0.05, vs. Control group (χ^2^ test).

**Table 2 cells-13-00308-t002:** Percentages of immunostained cells for various markers in EdU-single- and BrdU-single-labeled cells in the Purkinje cell layer of the cerebellar cortex.

	Anterior Lobe		Posterior Lobe	
	Control	VPA	Control	VPA
EdU-single labeled cells				
% of Pax6	33.3% (1/3)	0% (0/4)	12.5% (1/8)	0% (0/5)
% of NeuN	100% (3/3)	25.0% (1/4) *	100% (1/1)	100% (1/1)
% of PCNA	0% (0/2)	0% (0/3)	0% (0/3)	0% (0/3)
% of S100	0% (0/14)	10.0% (1/10)	11.1% (1/9)	0% (0/2)
% of BLBP	0% (0/2)	0% (0/7)	0% (0/9)	0% (0/6)
BrdU-single labeled cells				
% of Pax6	90.0% (9/10)	100% (25/25)	62.5% (10/16)	100% (29/29) **
% of NeuN	100% (1/1)	0% (0/11)	40.0% (2/5)	7.7% (1/13)
% of PCNA	0% (0/14)	31.3% (10/32)	19.0% (4/21)	41.0% (16/39)
% of S100	11.8% (2/17)	45.0% (9/20) *	0% (0/10)	23.8% (5/21) *
% of BLBP	0% (0/15)	0% (0/38)	0% (0/17)	0% (0/36)

Percentages are calculated by summing each labeled cell counted within all ROIs from Purkinje cell layer of cerebellar cortex from four ferrets. The number of each labeled cell for calculating the percentages is shown in parentheses. * *p* < 0.05, ** *p* < 0.01, vs. Control group (χ^2^ test).

**Table 3 cells-13-00308-t003:** Percentages of immunostained cells for various markers in EdU-single- and BrdU-single-labeled cells in the internal granular layer of the cerebellar cortex.

	Anterior Lobe		Posterior Lobe	
	Control	VPA	Control	VPA
EdU-single labeled cells				
% of Pax6	50.0% (2/5)	15.4% (2/13)	0% (0/8)	11.8% (2/17)
% of NeuN	80.0% (4/5)	55.6% (5/9)	33.3% (2/6)	33.3% (4/12)
% of PCNA	0% (0/11)	33.3% (4/12) *	11.1% (2/18)	0% (0/16)
% of S100	0% (0/12)	4.5% (1/22)	0% (0/16)	0% (0/34)
% of BLBP	0% (0/7)	0% (0/29)	12.5% (1/8)	9.5% (2/21)
BrdU-single labeled cells				
% of Pax6	85.9% (30/35)	94.7% (108/114)	79.1% (53/67)	94.3% (100/106) **
% of NeuN	33.3% (10/30)	44.6% (29/65)	38.2% (21/55)	57.5% (42/73) *
% of PCNA	4.39% (3/61)	17.9% (12/67) *	7.0% (7/100)	17.8% (12/129) *
% of S100	5.3% (4/75)	7.3% (7/96)	17.5% (11/63)	14.9% (15/101)
% of BLBP	2.6% (2/77)	15.7% (16/102) **	2.8% (2/71)	10.2% (10/98)

Percentages are calculated by summing each labeled cell counted within all ROIs from inner granular layer of the cerebellar cortex of four ferrets. The number of each labeled cell for calculating the percentages is shown in parentheses. * *p* < 0.05, ** *p* < 0.01 vs. Control group (χ^2^ test).

## Data Availability

The original contributions presented in the study are included in the article/Appendix A, further inquiries can be directed to the corresponding author.

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
