# Peer review of "Tracking of Internal Granular Progenitors Responding to Valproic Acid in the Cerebellar Cortex of Infant Ferrets"

_cells, 2024, doi:10.3390/cells13040308_

Round 1

Reviewer 1 Report

Comments and Suggestions for Authors

This work is an extension of a previous study conducted by the same authors in the cerebellum of the ferrets (Kamiya et al., 2023 Front in Neurosci). While their previous work demonstrated the effect of valproic acid (VPA) on neurogenesis immediately after exposure and identified a population of progenitors they called "internal granular progenitors" (IGPs), the current study focuses on a longer term outcome.

In particular, they investigated the fate of IGPs and their output on postnatal day 20 and found a notable change in the abundance of specific neuronal and glia cell types in the cerebellum. The manuscript is well written and the analyses are well presented and explained. However, as pointed out by the authors, direct comparisons of relative proportions of different cell types in the same tissue was not carried out. As such, some interpretations such as the relative production of EGP- vs IGP- derived neurons suffer from the lack of direct evidence.

Nevertheless, the findings of this study are important and represent a significant continuation of their previous conclusions. Their discoveries also inspire future research in understanding the functional significance of diverse populations of progenitors in the ferret cerebellum and their potential evolutionary benefits. 

There are however several points that should be clarified:

1) Given that the effect of VPA is dependent on the time and duration of exposure, it is important to clarify why VPA administration on days 6 and 7 were chosen in this study. For example, the duration of action of VPA in the brain and potential off-target effects upon IP injections should be discussed. 

2) Authors should also discuss or assess whether the decrease of NeuN positive neurons in the ML might be due to cell death. The assessment of cerebellar volume mentioned is likely not sensitive enough to detect changes at cellular level, particularly when it concerns only some cell types. Direct assessment of cell death would be more appropriate.

3) It is not clear how variations among different ferrets (biological replicates) are statistically assessed in this study. For the counting of positively labeled cells, it appears that the counts of all 4 ferrets in each experimental group are pooled to produce a percentage which is then compared between two groups. Given that the authors have counts from individual ferrets, it would be more statistically relevant to pool all counts from each ferrets individually and then perform appropriate statistical tests on the mean of 4 biological replicates. Please present counts as suggested or justify the use of pooled data. 

Author Response

We have addressed each of the points raised by the reviewers. Please find our responses below.

Reply to Reviewer 1:

This work is an extension of a previous study conducted by the same authors in the cerebellum of the ferrets (Kamiya et al., 2023 Front in Neurosci). While their previous work demonstrated the effect of valproic acid (VPA) on neurogenesis immediately after exposure and identified a population of progenitors they called "internal granular progenitors" (IGPs), the current study focuses on a longer term outcome.

In particular, they investigated the fate of IGPs and their output on postnatal day 20 and found a notable change in the abundance of specific neuronal and glia cell types in the cerebellum. The manuscript is well written and the analyses are well presented and explained. However, as pointed out by the authors, direct comparisons of relative proportions of different cell types in the same tissue was not carried out. As such, some interpretations such as the relative production of EGP- vs IGP- derived neurons suffer from the lack of direct evidence.

Nevertheless, the findings of this study are important and represent a significant continuation of their previous conclusions. Their discoveries also inspire future research in understanding the functional significance of diverse populations of progenitors in the ferret cerebellum and their potential evolutionary benefits.

There are however several points that should be clarified:

Point 1. Given that the effect of VPA is dependent on the time and duration of exposure, it is important to clarify why VPA administration on days 6 and 7 were chosen in this study. For example, the duration of action of VPA in the brain and potential off-target effects upon IP injections should be discussed.

Response: This study was an extension of the previous study (Kamiya et al. Front Neurosci 2023) and to determine the fate of IGPs in response to VPA during the cerebellar corticohistogenesis. Therefore, VPA was injected into the ferret infants on the same schedule at PDs 6 and 7 when the EGL began to expand. These contexts were mentioned in the Introduction (L50-54, marked in yellow).

Point 2: Authors should also discuss or assess whether the decrease of NeuN positive neurons in the ML might be due to cell death. The assessment of cerebellar volume mentioned is likely not sensitive enough to detect changes at cellular level, particularly when it concerns only some cell types. Direct assessment of cell death would be more appropriate.

Response: We thank the reviewer for the valuable comment. The percentage of NeuN immunostaining in BrdU+ cells in the ML was significantly lower in VPA-exposed ferrets than in control ferrets (Table 1). In contrast, the percentage of Pax6 immunostaining in BrdU+ cells in the ML was significantly greater in VPA-exposed ferrets (Table 2). Based on these findings, we concluded that NeuN expression was delayed (maturation of granular neurons) in VPA-exposed ferrets. This is mentioned in the Results (L304-306, marked in yellow). Furthermore, VPA enhanced anti-apoptotic factors (such as Bcl-2) in neurol progenitors (Tsai et al. J Mol Med, 2008; Song et al. Prog Neuropsychopharmacol Biol Psychiatry, 2012) and inhibits apoptosis (Nikolian et al., J Trauma Acute Care Surg, 2018). Therefore, cell death was considered to be unrelated to the reduced percentage of NeuN immunostaining in BrdU+ cells in VPA-exposed ferrets. We have mentioned this in the Discussion (L399-401, marked in yellow).

Point 3: It is not clear how variations among different ferrets (biological replicates) are statistically assessed in this study. For the counting of positively labeled cells, it appears that the counts of all 4 ferrets in each experimental group are pooled to produce a percentage which is then compared between two groups. Given that the authors have counts from individual ferrets, it would be more statistically relevant to pool all counts from each ferrets individually and then perform appropriate statistical tests on the mean of 4 biological replicates. Please present counts as suggested or justify the use of pooled data.

Response: The cell density evaluates the effect of VPA on particular cerebellar regions at the tissue level. Therefore, we used the averages and standard errors for each measurement per animals as data. In contrast, immunostained ratios of ErdU+ or BdU cells for various antigens evaluate the cellular responses of progenitors (the direction and degree of differentiation) to VPA. Therefore, pooled data were used for this analysis.

Reviewer 2 Report

Comments and Suggestions for Authors

The manuscript by Kamiya et al. is well-written and obtained data are presented in tables and diagrams in a very comprehensive and clear way. Their findings are very interesting and sought to examine the role of IGPs as a new progenitor source for the generation of new granule cells and Bergman glia in the cerebellum in the healthy and VPA-affected brain. However, some points should be re-examined before considered for publication:

1.       I believe that a brief description of the structure of the three-layered cerebellar cortex and the types of neurons found in each layer should be added in the introduction.

2.       The methodology used for section selection and staining is poorly described. The authors should clearly state how many sections were immunostained and counted from each animal? Which was the interval in μm between serial sections? Moreover, every selected section was stained against BrdU or EdU and one marker of interest (ie Pax6, NeuN etc)? in tables 1-3 severe discrepancies arise in the total number of BrdU+ or EdU+ cells found in each section. For example in one section only 1 BrdU+ cell was counted whereas in other sections up to 39 BrdU+ cells were counted. This is due to the fact that sections from different cerebellar levels (according to bregma), that contain a different number of neuronal cells, were selected and compared or other?

3.       In lines 191-193 the authors state that EdU and BrdU labeling was not found in Purkinje cells of the PCL nor in interneurons of the ML. However in tables 1 and 2 it seems that the vast majority of PCL and ML BrdU or EdU+ cells express markers of the neuronal lineage (NeuN, Pax6 etch) and not astroglial or progenitor cell markers. How do the authors explain that?

4.       Expression of Pax6 is evident during the proliferation and early differentiation period of a newborn neurons. In this experiment ferrets were euthanized at 13d or 15d after BrdU or EdU injection respectively, where newborn neuronal cells should have overcome their differentiation period and reached their early maturation period. How do the authors explain the extremely high coexpression percentages of proliferation markers and Pax6 at these days?

5.       S100 is a marker of fully mature and not differentiated or immature astrocytes. The mean maturation time for a new astrocyte is almost 21d ie in the hippocampus. How do the authors explain the expression of s100 by newborn Bergmann glia cells as soon as 13d or 15d after their genesis?

6.       How do the authors explain the significant differences found between the anterior and posterior cerebellar lobe? This should be discussed in the manuscript.

Author Response

We have addressed each of the points raised by the reviewers. Please find our responses below.

Reply to Reviewer 2:

The manuscript by Kamiya et al. is well-written and obtained data are presented in tables and diagrams in a very comprehensive and clear way. Their findings are very interesting and sought to examine the role of IGPs as a new progenitor source for the generation of new granule cells and Bergman glia in the cerebellum in the healthy and VPA-affected brain. However, some points should be re-examined before considered for publication:

Point 1. I believe that a brief description of the structure of the three-layered cerebellar cortex and the types of neurons found in each layer should be added in the introduction.

Response: Following the reviewer’s suggestion, we have mentioned the structure of the three-layered cerebellar cortex and neuronal types observed in each layer in the Introduction (L32-38, marked in yellow).

Point 2. The methodology used for section selection and staining is poorly described. The authors should clearly state how many sections were immunostained and counted from each animal? Which was the interval in μm between serial sections? Moreover, every selected section was stained against BrdU or EdU and one marker of interest (ie Pax6, NeuN etc)? in tables 1-3 severe discrepancies arise in the total number of BrdU+ or EdU+ cells found in each section. For example in one section only 1 BrdU+ cell was counted whereas in other sections up to 39 BrdU+ cells were counted. This is due to the fact that sections from different cerebellar levels (according to bregma), that contain a different number of neuronal cells, were selected and compared or other?

Response: Six serial sections (100 μm-thickness, no gaps between adjacent sections) were prepared from the cerebellar vermis around the midsagittal plane. One section was stained with hematoxylin for gross histological evaluation. The remaining five sections were used for EdU and BrdU detections with immunostaining for five different antigens. This is mentioned in the Materials and Methods (L85-88, marked in yellow).

Point 3. In lines 191-193 the authors state that EdU and BrdU labeling was not found in Purkinje cells of the PCL nor in interneurons of the ML. However in tables 1 and 2 it seems that the vast majority of PCL and ML BrdU or EdU+ cells express markers of the neuronal lineage (NeuN, Pax6 etch) and not astroglial or progenitor cell markers. How do the authors explain that?

Responses: Thanks for your comment. As mentioned in the Results, Pax6 is expressed in migrating/differentiating granular neurons and IGPs in the developing cerebellum (Chung et al. Anat Cell Biol, 2010; Sawada and Kamiya, Congenit Anom, 2023) (L225-226, marked in yellow). Furthermore, NeuN, a marker of mature neurons, is expressed in granular neurons but not in other cerebellar cortical neurons such as Purkinje cells and ML interneurons (Wolf et al. J Histochem Cytochem 1996; Weyer et al. J Neurosci Res 2003) (L297-298, marked in yellow). Therefore, we excluded the possibility that Pax6+ and NeuN+ cells were Purkinje cells and ML interneurons.

Point 4. Expression of Pax6 is evident during the proliferation and early differentiation period of a newborn neurons. In this experiment ferrets were euthanized at 13d or 15d after BrdU or EdU injection respectively, where newborn neuronal cells should have overcome their differentiation period and reached their early maturation period. How do the authors explain the extremely high coexpression percentages of proliferation markers and Pax6 at these days?

Responses: Pax6 was expressed in migrating/differentiating granular neurons and/or IGPs in the developing cerebellum (Chung et al. Anat Cell Biol, 2010; Sawada and Kamiya, Congenit Anom, 2023). We consider that many cells proliferated on PDs 5 (EdU+) and 7 (BrdU+) are immature granular neurons or undifferentiated IGPs on PD 20.

Point 5. S100 is a marker of fully mature and not differentiated or immature astrocytes. The mean maturation time for a new astrocyte is almost 21d ie in the hippocampus. How do the authors explain the expression of s100 by newborn Bergmann glia cells as soon as 13d or 15d after their genesis?

Responses: The S100 gene was expressed largely in immature Bergman glia in the mouse cerebellum from embryonic day 13.5 to PD 3 (Vives et al. J Comp Neurol, 2004). S100+ cells were found in the IGL adjacent to the PCL in the ferret cerebellum on PD 7 (Kamiya et al. Front Neurosci, 2023) and some of them were co-stained with anti-PCNA on PD 20 (Figure S1a). Therefore, we consider that S100 was expressed in differentiating/immature Bergmann glia as well as mature Bergmann glia in the ferret cerebellar cortex. This context is mentioned in the Results (L329-334, marked in yellow). In this study, BLBP was used as a marker for mature Bergmann glia (L345, marked in yellow).

Point 6. How do the authors explain the significant differences found between the anterior and posterior cerebellar lobe? This should be discussed in the manuscript.

Responses: In this study, cell densities were statistically evaluated using a repeated-measures two-way ANOVA with the cerebellar lobes (anterior and posterior) and groups (VPA-exposed and control) as factors. When a significant effect on the interaction between cerebellar lobes and groups is revealed by two-way ANOVA, the post-hoc testing indicates that the difference in the cell density between the two groups is specific to the cerebellar lobes (anterior or posterior or lobes). On the other hand, the two-way ANOVA reveals a significant effect on groups, but not on the cerebellar lobe × group interaction; the difference between the two groups is not specific to the cerebellar lobes. In this case, the post-hoc testing indicates a significant difference between the groups only in the anterior lobe. The difference between the groups in the posterior lobe is interpreted to have a tendency similar to that in the anterior lobe. These interpretations have been mentioned in the manuscript (L192-195, L377-382, marked in yellow).